# A Randomized, Double-Blind, Placebo-Controlled, Multicenter Study to Evaluate the Safety and Efficacy of ThymoQuinone Formula (TQF) for Treating Outpatient SARS-CoV-2

**DOI:** 10.3390/pathogens11050551

**Published:** 2022-05-07

**Authors:** Hassan Bencheqroun, Yasir Ahmed, Mehmet Kocak, Enrique Villa, Cesar Barrera, Mariya Mohiuddin, Raul Fortunet, Emmanuel Iyoha, Deborah Bates, Chinedu Okpalor, Ola Agbosasa, Karim Mohammed, Stephen Pondell, Amr Mohamed, Yehia I. Mohamed, Betul Gok Yavuz, Mohamed O. Kaseb, Osama O. Kasseb, Michelle York Gocio, Peter Tsu-Man Tu, Dan Li, Jianming Lu, Abdulhafez Selim, Qing Ma, Ahmed O. Kaseb

**Affiliations:** 1RESPIRE Clinical Research, Palm Springs, CA 92262, USA; r.fortunet@respireresearch.com; 2United Memorial Medical Center, Department of Research and Development, Houston, TX 77091, USA; yasirahmed.med@gmail.com (Y.A.); cbarrera@ummc.care (C.B.); mariya@ummc.care (M.M.); 3Department of Biostatistics and Medical Informatics, International School of Medicine, Istanbul Medipol University, 34810 Istanbul, Turkey; mehmet.kocaktas@gmail.com; 4L&A Morales Healthcare, Inc., Miami, FL 33012, USA; evilla@moraleshealthcare.com; 5Tranquil Clinical and Research Consulting Services, Houston, TX 77598, USA; emmanueli@tranquilconsulting.com (E.I.); deborahb@tranquilconsulting.com (D.B.); chineduo@tranquilconsulting.com (C.O.); olaa@tranquilconsulting.com (O.A.); karim@tranquilconsulting.com (K.M.); 6Chemistry, Manufacturing and Controls Department, Novatek Pharmaceuticals, Inc., Houston, TX 77054, USA; spondell@ibtsolutions.us; 7UH Seidman Cancer Center, Case Western Reserve University, Cleveland, OH 44106, USA; amr.mohamed@uhhospitals.org; 8Department of Gastrointestinal Medical Oncology, The University of Texas MD Anderson Cancer Center, Houston, TX 77030, USA; yimohamed@mdanderson.org (Y.I.M.); bgok@mdanderson.org (B.G.Y.); 9Novatek Pharmaceuticals, Inc., Houston, TX 77598, USA; mkaseb1@gmail.com (M.O.K.); okasseb@hotmail.com (O.O.K.); mgocio@novatekpharmaceuticals.com (M.Y.G.); 10Law Offices of Peter Tu LLC, Plainsboro, NJ 08536, USA; pt22064@comcast.net; 11Department of Hematopoietic Biology and Malignancy, University of Texas MD Anderson Cancer Center, Houston, TX 77030, USA; danli@mdanderson.org (D.L.); qma@mdanderson.org (Q.M.); 12Department of Biochemistry and Molecular and Cellular Biology, Georgetown University School of Medicine, Washington, DC 20007, USA; jimmy_lu@codexbio.com; 13Codex BioSolutions Inc., Rockville, MD 20852, USA; 14Philadelphia College of Osteopathic Medicine (PCOM), Philadelphia, PA 19131, USA; abdu94@yahoo.com

**Keywords:** COVID-19, TQ Formula, pandemic, coronavirus, SARS-CoV-2

## Abstract

There is an urgent need for an oral drug for the treatment of mild to moderate outpatient SARS-CoV-2. Our preclinical and clinical study’s aim was to determine the safety and preliminary efficacy of oral TQ Formula (TQF), in the treatment of outpatient SARS-CoV-2. In a double-blind, placebo-controlled phase 2 trial, we randomly assigned (1:1 ratio) non-hospitalized, adult (>18 years), symptomatic SARS-CoV-2 patients to receive oral TQF or placebo. The primary endpoints were safety and the median time-to-sustained-clinical-response (SCR). SCR was 6 days in the TQF arm vs. 8 days in the placebo arm (*p* = 0.77), and 5 days in the TQF arm vs. 7.5 days in the placebo arm in the high-risk cohort, HR 1.55 (95% CI: 0.70, 3.43, *p* = 0.25). No significant difference was found in the rate of AEs (*p* = 0.16). TQF led to a significantly faster decline in the total symptom burden (TSB) (*p* < 0.001), and a significant increase in cytotoxic CD8^+^ (*p* = 0.042) and helper CD4^+^ (*p* = 0.042) central memory T lymphocytes. TQF exhibited an in vitro inhibitory effect on the entry of five SARS-CoV-2 variants. TQF was well-tolerated. While the median time-to-SCR did not reach statistical significance; it was shorter in the TQF arm and preclinical/clinical signals of TQF activity across multiple endpoints were significant. Therefore, a confirmatory study is planned.

## 1. Introduction

To date, there has been no full approval by US FDA for an oral drug to shorten disease course in mild to moderate outpatient SARS-CoV-2. Traditional herbal medicines and purified natural products represent an attractive therapeutic option given their known safety and tolerability.

Nigella sativa, commonly known as black seed, has been shown in several studies to exert antiviral and anti-inflammatory activities [1,2,3,4,5,6]. Emerging data suggest that its active ingredient, thymoquinone (TQ), may block SARS-CoV-2 cellular entry, possibly via blocking the angiotensin-converting enzyme 2 (ACE2) receptor [7]. Nigella sativa was associated with faster recovery of symptoms in patients with a mild COVID-19 infection in a recent study [8]. TQ Formula (TQF) is a patent pending formulation which is the first-ever enteric-coated capsule, derived from Nigella Sativa oil, characterized and manufactured under Good Manufacturing Practices (GMP) with a specific tight TQ concentration of 1.7%.

Our preclinical and clinical studies sought to evaluate the safety and clinical efficacy of TQF through a randomized, double-blind, placebo-controlled, multicenter phase II, which utilized an earlier generation of TQF as the investigational intervention.

## 2. Results

### Clinical Study Results

Between 27 May 2021, and 27 September 2021; 79 patients were screened for the randomized trial, of whom 24 were ineligible. All 55 randomized patients were analyzed based on an intent-to-treat analysis. The same analyses were repeated based on the per-protocol dataset requiring 100% compliance throughout the therapy as well as the full analysis dataset requiring that patients receive at least one dose of the study drug. The main study findings were consistent across these protocol-defined analysis sets. Figure 1 shows the trial participation profile.

Patient demographics are shown in Table 1. The mean age was 45 years, 24 patients (43.64%) were males, 25 (45.45%) were white, 22 (40%) were Hispanic or Latino, 7 (12.73%) were black, and 1 (1.82%) was Asian. Co-morbidities included obesity (38.18%), hypertension (40%), and diabetes mellitus (18.18%). A total of 9/55 (16.4%) of patients were fully vaccinated.

For the primary safety endpoint, we compared AEs for the 52 patients who took any dose of TQF or placebo. Three of the twenty-nine (10.3%) patients treated with TQF experienced a total of three treatment-related AEs (two mild and one moderate), while 6 of 23 (26.1%) patients treated with placebo experienced a total of nine treatment-related AEs (eight mild and one moderate) (*p* = 0.16). There was a total of 15 AE episodes (5 in the treatment arm and 10 in the placebo arm) in 11 patients (4 in the TQF arm and 7 in the placebo arm), and one SAE reported in the placebo arm in the form of hypoxia and pneumonia that led to hospitalization. These results indicated that TQF was safe and tolerable. Furthermore, for the sustained clinical response (SCR) primary endpoint; in applying an intent-to-treat analysis, a shorter but not significant median time-to-SCR was seen in the TQF arm (6 days; 95% CI: 4–14) than in the placebo arm (8 days; 95% CI: 6–11), (*p* = 0.77, HR = 1.10, 95% CI: 0.58–2.07, Figure 2A), not surprisingly because the original power calculation of the study was assuming that the median time-to-SCR in the placebo arm would be around 16 days. Importantly, among high-risk patients, an even shorter median time-to-SCR was seen in the TQF arm (5 days; 95% CI: 3.7) than in the placebo arm (7.5 days; 95% CI: 3, non-estimable), (HR = 1.55), 95% CI: 0.70, 3.43, *p* = 0.25), Figure 2B).

For the secondary endpoints one and two, the VL distribution did not differ between the arms at baseline and on day 7, but on day 14, patients treated with TQF had a suggestively lower VL (Table 2), due to a higher but not statistically significant (*p* = 0.22) number of SARS-CoV-2-negative cases on day 14 in the TQF arm (16 of 21 patients; 76.2%) compared with the placebo arm (11 of 19 patients; 57.9%). Longitudinal modeling also showed a similar result, but with a sharper decline trend of VL in the TQF arm than in the placebo arm, *p* = 0.146, still not meeting statistical significance (Figure 2C).

The secondary endpoint three summarized the symptom burden overall and in each subdomain from day 1 through day 14. TQF led to a statistically significant faster decline in the overall symptom burden TSB (*p* < 0.0001), as well as three out of the six subdomains: Throat (*p* = 0.0003) and Body/Systemic (*p* = 0.0011) symptom burdens even under a more stringent significance threshold of 0.007 with Bonferroni multiple testing correction (Figure 2D and Appendix A). Symptom burdens not significant at 0.007 were Chest/Respiratory (*p* = 0.024), Gastrointestinal (*p* = 0.124), and Smell/Taste (*p* = 0.032) domains, although they declined suggestively much faster with TQF (Appendix A).

For the secondary endpoint 4; there was no significant association between VL and TSB (*p* = 0.14) controlling for time, and no interaction between time and VL (*p* = 0.77); in addition, there was no differential association by treatment arm, either (*p* = 0.17). The same conclusion was valid for all subdomains as well.

For the exploratory endpoint of flow cytometry assessments for effector T Cells, they were available for 49 participants (27 in the TQF arm and 22 in the placebo arm). In this analysis, only the assessment for effector T lymphocytes was completed, since testing for pharmacokinetics, inflammatory cytokines, and coagulation factors were not available by the same vendors at the time of the VL and flow cytometry analyses. Using flow cytometry for various T cell surface markers, helper CD4^+^ increased significantly (*p* = 0.042) as did cytotoxic CD8^+^ (*p* = 0.042) T lymphocytes with native/central memory phenotype (CD45RA^+^CCR7^+^) in patients treated with TQF compared to placebo (Figure 2E, Appendix A, and a results summary in Table 3).

## 3. Discussion

Our study reached its primary safety endpoint. TQF was predictably safe and well-tolerated. Our other co-primary endpoint, the median time-to-sustained-clinical-response (SCR), was shorter by 2 days in the overall population, and 2.5 days in high-risk patients. These differences did not reach statistical significance, explained by the more abbreviated recovery time of 8 days in the placebo arm (Predefined 16 days, based on early pandemic data in 2020 from CDC) [9]. However, our TQF arm has been validated (Predefined 6 days recovery). With respect to high-risk patients; 32/55 (58%) of our population matched this definition. In those, median time-to-SCR was shorter (5 vs. 7.5 days in placebo; HR 1.55 (95% CI: 0.70, 3.43, *p* = 0.29). Future larger TQF studies are needed to confirm this strong trend in high-risk patients. Moreover, we maintain that a 2–2.5 days difference is still clinically meaningful, especially to the patients.

Our study secondary outcomes included individual and total symptom burden (TSB). The difference between the two arms was a statistically significant (faster) decline in TSB (*p* ≤ 0.0001), throat (*p* = 1.0003), and body/systemic symptom (*p* = 0.0011). This is very relevant to the Omicron SARS-CoV-2 surge, which causes a higher incidence of upper respiratory and bodily symptoms. Symptom burdens not significant at 0.007 were Chest/Respiratory (*p* = 0.024), Gastrointestinal (*p* = 0.124), and Smell/Taste (*p* = 0.032) domains, although they declined suggestively much faster with TQF therapy. FLU-PRO plus is a reliable self-administered SARS-CoV-2 symptom scoring system that has been validated and suggested as a tool to compare results from different studies of viral pulmonary infections [10]. Indeed, with the advent of multiple trials for SARS-CoV-2 therapeutics, patient-reported quality of life as a core outcome measure was recommended by the Global SARS-CoV-2 Core Outcomes Set (SARS-CoV-2 COS) initiative [10]. Our paper validated the Modified FLU-PRO Plus questionnaire as a SARS-CoV-2 symptom tool as it focuses on what is clinically relevant to the patient. Viral load (VL) as a secondary endpoint analysis displayed a trend of a faster, though admittedly not significant, decline in VL (*p* = 0.146) after 14 days of TQF treatment (Figure 2C), which could be related, at least partially, to the limitation of our VL data reporting an upper threshold ceiling of 25,000, and only at three time points, which in turn limited the modeling ability to describe its longitudinal change.

The exploratory analysis of effect T lymphocytes showed a significant increase in cytotoxic CD8^+^ T lymphocytes (*p* = 0.042) and helper CD4^+^ T lymphocytes (*p* = 0.042) with native/central memory phenotype (CD45RA^+^CCR7^+^), from baseline to day 14, in TQF arm as compared to placebo. This suggests that TQF supports the recovery of the immune system against SARS-CoV-2, which likely helps prevent the progression to severe SARS-CoV-2 [11,12,13,14,15]. We also posit that TQF might directly prevent the overall T cell exhaustion and promote SARS-CoV-2-specific T cell proliferation.

Notably, our molecular docking models and in vitro experiments confirmed TQF anti-SARS-CoV-2 inhibitory effects against five SARS-CoV-2 variants, including Omicron, likely through viral entry antagonism at the ACE-2 receptor level, combined with the in vivo immunomodulatory activity and the potential anti-inflammatory actions of TQF may in aggregate explain the positive signals noted in this clinical trial (Figure 2F and Appendix A).

Our study has several strengths, including its novel drug formulation, randomized, double-blind, placebo-controlled, and multicenter design, TQF activity across multiple preclinical and clinical endpoints, and the novel TQF effect on specific SARS-CoV-2-CD4^+^/CD8^+^ T cells; CD45RA^+^CCR7^+^-expressing T-lymphocytes. Another strength is our more diverse minority population representation which was higher than the average US population in our sample, with 29/55 patients (52.7%) being Hispanic and/or African-American. These groups are reported to have a higher incidence of morbidity and mortality (possibly due to ACE2 expression in tissues [16,17,18]), are more prone to vaccine hesitancy, and are more open to naturally derived treatments. Furthermore, our strong in vitro data of TQF’s anti-SARS-CoV-2 effects suggested blockage of cellular entry, which makes this a variant-independent potential supportive and ubiquitous therapy (see Appendix A). Although some of our endpoints did not reach statistical significance (time-to-SCR and VL), the trend towards benefit in the treatment arm as compared to the placebo arm is encouraging and warrants a larger sample study. The current study has some limitations, including the small sample size for some of its endpoints, selected mostly to prove safety, and allow proof of concept in a randomized design of the study. Another limitation is related to the unavailability of inflammatory and coagulation markers by the same vendors at the time of the VL and flow cytometry analyses and the VL measurements ceiling cap effect of 25,000 on our VL analysis.

Although our SCR primary endpoint was shorter in the TQF arm, it did not reach statistical significance. However, TQF showed safety, preclinical and clinical signals of activity across multiple secondary and exploratory endpoints in this randomized study. TQF reduced symptoms burden, mostly in intensity but suggestively in time as well (though clinically but not statistically), which remains the most salient core outcome relevant to patients with mild-to-moderate SARS-CoV-2. Furthermore, the significant increase in central memory cells in the TQF arm suggests faster immune recovery and potential long-term benefit in reduction of re-infection, hospitalization, and deaths. Based on the totality of the data from our study, a larger phase three study is planned, and Sponsor is actively engaging with the US and global regulatory bodies to determine potential next steps which may lead to its full approval to enhance clinical recovery and prevent severe infections in outpatient mild-to-moderate SARS-CoV-2.

## 4. Materials and Methods

### 4.1. Clinical Study Design

BOSS-001 trial (ClinicalTrials.gov Identifier: NCT04914377) was a national randomized (1:1), double-blind, placebo-controlled phase 2 study to assess the safety and efficacy of the oral drug TQF (500 mg capsules, 3 capsules, twice a day for 14 days) versus placebo capsules in treating patients who had tested positive for SARS-CoV-2 in the outpatient setting. The trial was conducted at three sites in the United States; RESPIRE Research, LLC, Palm Springs, CA, USA; United Memorial Medical Center, Houston, TX, USA; and L&A Morales Clinic, Miami, FL, USA. The trial protocol was approved by a centralized institutional review board, WCG IRB, Puyallup, WA, USA, and overseen by a contract research organization, Tranquil Clinical Research, Houston, TX.

### 4.2. Participants

This investigational new drug (IND) study received a “Study May Proceed” letter from the FDA in April 2021. All symptomatic patients with SARS-CoV-2 at all study sites were screened for trial eligibility. Key inclusion criteria included age ≥18 years, positive SARS-CoV-2 within the last 3 days and confirmed with an RT-PCR test at baseline, and presentation with mild-to-moderate clinical symptoms of SARS-CoV-2 infection, defined as a score of ≥3 on a minimum of two symptoms on the Modified FLU-PRO Plus patient-reported measures Questionnaire. Written informed consent was obtained from each patient.

### 4.3. Randomization and Masking

Participants were randomized (1:1) using block randomization with a block size of 4, stratified by the participating sites. Randomization and data collection were both completed using the Electronic Data Capture (EDC) system. The investigational and placebo drugs were indistinguishable and GMP-manufactured. All subjects, investigators, and study personnel, including the primary biostatistician, were blinded to the treatment assignment. An independent unblinded statistician generated the randomization list but did not otherwise participate in the study procedures or data analysis.

### 4.4. Procedures

Patients were assessed on days 1, 4, 7, 10, and 14, and a final study follow-up on day 21, with a safety follow-up phone call on day 45. SARS-CoV-2 symptoms were self-reported daily from randomization to the final study visit, using the Modified FLU-PRO Plus questionnaire. FLU-PRO plus symptoms; individual, domain, and total scores were calculated. Nasopharyngeal swab samples (red-top tubes) for a quantitative viral load (VL) measured by RT-PCR, and blood samples (EDTA violet-top tubes) for flow cytometry were collected on days 1, 7, and 14 and shipped ambient to be tested at Cerba Research, Lake Success, NY, USA.

### 4.5. Outcomes

The primary endpoints were: (1) to evaluate the safety and tolerability of TQF, by assessing the number of overall adverse events (AEs), treatment-related AEs, and serious AEs (SAEs), including hospitalizations and (2) to evaluate if treatment with TQF can significantly reduce the median time-to-sustained clinical response (SCR) compared to placebo; SCR was defined as a reduction of scores to ≤2 on all symptoms of the FLU-PRO Plus if they remained so for at least three days.

The secondary endpoints were: (1) to compare the VL profile over time, from day 1 through day 14) between TQF and placebo arms, (2) to compare the percentage of RT-PCR negative/undetectable (i.e., viral clearance) on day 7 and day 14 between TQF and placebo arms, and (3) to assess the total symptom burden (TSB) through comparing the duration and severity of symptoms over time, from day 1 through day 14; in terms of the individual symptoms, total score and the sub-domain scores (namely Nose, Throat, Eyes, Chest/Respiratory, Gastrointestinal, Body/Systemic, Taste/Smell), between TQF and placebo arms, and (4) To determine whether there exists an association between the VL and the symptom severity by study arm and whether any such associations change over time.

Our exploratory endpoint included comparing treatment effects on immune cells on days 1, 7, and 14 between TQF and placebo arms.

### 4.6. Statistical Analysis

The trial was designed to enroll up to 60 patients equally randomized between the two arms of the study. The safety monitoring phase of the study consisted of up to 12 patients, whose adverse event profiles were monitored continuously daily to halt the study if ≥3 out of the initial 12 patients experienced grade 3 or higher adverse events attributable to the study regimen and not resolved through symptomatic treatments within 48 h. The primary statistical analysis was based on an intent-to-treat. An additional per-protocol analysis dataset was defined as the participants who completed all visits up to and including day 14, were compliant with the study medication, and did not have any major protocol violations. Major protocol violations were captured prior to breaking the blind. Additional subgroup analysis performed for high-risk patients (at least one risk factor; age ≥ 60, hypertension, diabetes, obesity, chronic cardiopulmonary disease, or auto-immune disease, selected similar to recent randomized studies [19,20,21,22,23,24].

Median time-to-SCR was computed from the date of randomization to the date when all 24 symptoms were at level 2 or below and had remained so, for at least 3 days. Based on the median time-to-SCR reported by the Center for Disease Control and Prevention (CDC) at the beginning of the pandemic in 2020 [8], the expected median time-to-recovery in the placebo arm was approximately 16 days for mild to moderate outpatient SARS-CoV-2. With a sample size of 20 in each arm, a difference in the median time-to-SCR of 10 days between the intervention arm and the placebo arm would be able to be detected with 81.5% statistical power with a 5% Type-1 error rate. With an attrition expectation of around 20%, the planned accrual was 26 patients in each arm and potentially up to 30 patients. Patients who did not achieve SCR were censored on the last date for which the FLU-PRO Plus data were available. SCR-free survival was computed by the Kaplan–Meier product-limit method and compared between the two arms of the study by the log-rank test. Median time-to-SCR was estimated through the Kaplan–Meier product-limit method.

In comparing viral load (VL) distribution between the two arms of the study cross-sectionally, the Wilcoxon–Mann–Whitney test, a non-parametric alternative to the two-sample *t*-test, was used, where any measurement provided as ‘>25,000’ was conservatively considered as 25,000 and the VL measurements of SARS-CoV-2-negative samples were considered as zero. For longitudinal modeling of VL, a random coefficient modeling approach through the SAS MIXED procedure was used with individual specific intercepts and slope estimates.

The same approach was used for longitudinal individual symptom burden assessments, where both linear and quadratic changes over time were added to the model. We added symptom scores to obtain a measure of total symptom burden from days 1–14; similarly, subdomain scores were obtained as unweighted additions of relevant symptom scores. In all these longitudinal models, the primary interest was to investigate whether there was a significant change in these markers over time and whether such change depended on the treatment arm.

For the T lymphocytes subsets, compared on day 1, day 7, and day 14, the above-mentioned non-parametric testing approach was also used. As the flow-cytometry analysis was exploratory in nature, no multiplicity correction was carried out. Consequently, the widths of the intervals have not been adjusted for multiplicity and the inferences drawn may not be reproducible. All analyses were conducted using SAS version 9.4.

## Figures and Tables

**Figure 1 pathogens-11-00551-f001:**
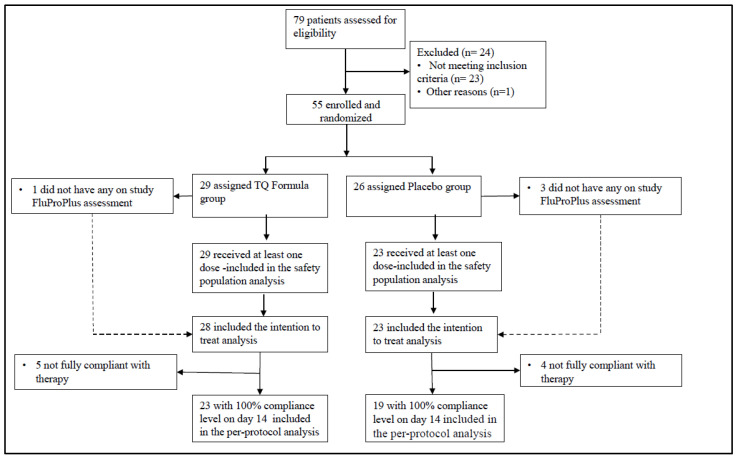
Consort diagram.

**Figure 2 pathogens-11-00551-f002:**
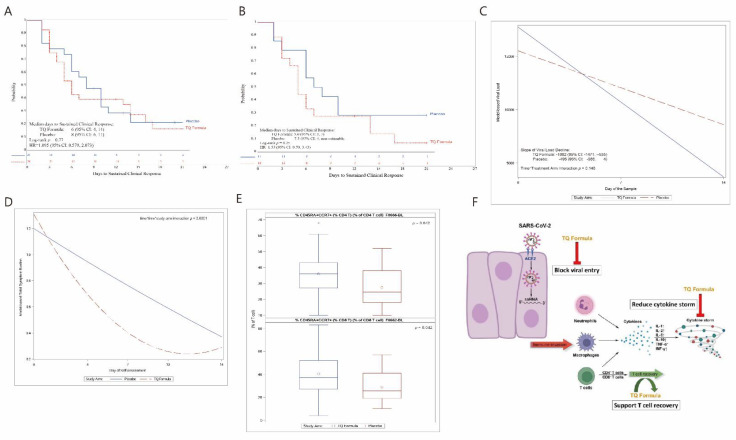
(**A**) Time-to-sustained-clinical-response (SCR) distribution by Kaplan–Meier curve. Shown are Kaplan–Meier estimates of SCR for TQF (blue) and placebo (red) arms. Stratified hazard ratios for SCR are reported, along with *p* values. CI denotes confidence interval. (**B**) Time-to-SCR distribution by Kaplan–Meier curve for high-risk patients. Shown are Kaplan–Meier estimates of SCR for TQF (blue) and placebo (red) arms. Stratified hazard ratios for SCR are reported, along with *p* values. (**C**) Viral load distribution over time by treatment arm. Shown are the viral load distribution from day 0 to day 14 for TQF (blue) and placebo (red) arms. (**D**) Model-based change of total symptom burden (TSB) change by study arm. Shown are the model-based change of TSB, defined as the duration and severity of symptoms over time, for TQF (blue) and placebo (red) arms. (**E**) Comparison of T cells between the treatment arms. Shown are comparison of percentage of CD45RA^+^CCR7^+^ in CD4 and CD8 T cells on day 14 between TQF (blue) and placebo (red) arms. (**F**) Potential mechanism of action of TQ Formula. TQ Formula may be effective treating COVID-19 via inhibiting viral entry through ACE2 blockage, reducing cytokine storm and promoting T cell recovery.

**Table 1 pathogens-11-00551-t001:** Patients Demographics and Comorbidities.

		All	TQ Formula	Placebo
		N (%)	N (%)	N (%)
Sex				
Female		31 (56.36)	16 (55.17)	15 (57.69)
Male		24 (43.64)	13 (44.83)	11 (42.31)
Race/Ethnicity				
Asian	Non-Hispanic/Latino	1 (1.82)	1 (3.45)	
Black or African American	Hispanic/Latino	3 (5.45)	1 (3.45)	2 (7.69)
	Non-Hispanic/Latino	4 (7.27)	1 (3.45)	3 (11.54)
White	Hispanic/Latino	20 (36.36)	11 (37.93)	9 (34.62)
	Non-Hispanic/Latino	5 (9.09)	2 (6.9)	3 (11.54)
Others	Hispanic/Latino	22 (40.00)	13 (44.83)	9 (34.62)
Age				
Years (Mean ± SD)		45.69 ± 17.35	45.48 ± 19.29	45.92 ± 15.27
≤55 years		39 (70.91)	21 (72.41)	18 (69.23)
>55 years		16 (29.09)	8 (27.5)	8 (30.76)
Common Comorbidities				
Diabetes Mellitus 2		10 (18.18)	5 (17.24)	5 (19.23)
Hypertension		22 (40.00)	11 (37.93)	11 (42.31)
BMI				
Underweight		1 (1.82)	1 (3.45)	
Normal weight		9 (16.36)	2 (6.90)	7 (26.92)
Overweight		24 (43.64)	14 (48.28)	10 (38.46)
Obese		21 (38.18)	12 (41.38)	9 (34.62)
SARS-CoV-2 Vaccination *		9 (16.36)	7 (24.13)	2 (7.69)

* 8 patients were fully vaccinated, and 1 patient have received one dose of vaccine at baseline.

**Table 2 pathogens-11-00551-t002:** RT-PCR Viral Load Distribution.

		N	Min	Q1	Median	Q3	Max	Mean	SD	*p*-Value *
Viral Load Day 0	Placebo	23	59	1683	25,000	25,000	25,000	15,418	11,465	0.69
	TQ Formula	25	87	9575	25,000	25,000	25,000	17,819	9991	
Viral Load Day 7	Placebo	18	0	0	12,282	25,000	25,000	12,479	12,539	0.81
	TQ Formula	23	0	0	216	25,000	25,000	10,674	12,407	
Viral Load Day 14	Placebo	19	0	0	0	25,000	25,000	9056	11,907	0.18
	TQ Formula	21	0	0	0	0	25,000	4274	8513	

* Based on Wilcoxon–Mann–Whitney tests comparing Blackseed Oil with placebo arms. Longitudinal modeling utilizing within patient change of viral load from baseline to day 7 and day 14 also showed a similar indication with sharper decline of viral load in TQF arm versus placebo arm * viral load measurements were not available for some patients due to sample not being collected or indeterminate RT-PCR results. Abbreviations: N, number; Q1, first quartile; Q3, third quartile; SD, standard deviation; min, minimum; max, maximum.

**Table 3 pathogens-11-00551-t003:** Summary of objectives, endpoints and results.

Objectives/Purpose	Endpoints/Outcome Measures	Justification and Results for Endpoints
**Primary**		
To evaluate if treatment with 3 g TQ Formula (500 mg per capsule, 3 capsules BID) given orally on outpatient basis can significantly reduce median time-to-sustained-clinical-response compared to placebo in participants with COVID-19 infection treated in the outpatient setting.	Measurement of the difference in median time-to-sustained-clinical-response in participants taking 3 g TQ Formula (500 mg per capsule, 3 capsules BID) versus participants taking placebo. Sustained clinical response is defined as a reduction of scores to ≤2 on all symptoms of the Modified FLU-PRO Plus.	A reduction in time-to-sustained-clinical-response is a direct measure of treatment effectiveness. Results: median time-to-SCR of 6 days in TQ Formula arm (95% CI: 4.14) versus 8 days in placebo arm (95% CI: 6.11). (*p* = 0.77).
To evaluate the safety and tolerability of TQ Formula (500 mg oral capsule, 3 capsules BID) when given to participants with COVID-19 infection.	Number of overall adverse events, related adverse reactions, and hospitalizations reported in participants taking 3 g TQ Formula (500 mg per capsule, 3 capsules BID) versus participants taking placebo. All AEs/SAEs will be captured throughout the study as per schedule of assessments.	Comparing the number of AEs and SAEs and any relationship to IP is important for determining the safety profile of TQ Formula. Results: TQF arm had no SAEs and had 50% less AE incidence compared to placebo while this difference was not significant (*p* = 0.16).
**Secondary**		
To compare the viral load profile over time (from baseline through to day 14) between treatment with 3 g TQ Formula (500 mg per capsule, 3 capsules BID) given orally on outpatient basis and placebo in participants with COVID-19 infection.	Measurement of change in quantitative viral load from baseline, day 7, and day 14 using RT-PCR in participants taking 3 g TQ Formula (500 mg per capsule, 3 capsules BID) versus participants taking placebo with COVID-19 infection.	Faster decline in viral load is hypothesized to lead to faster recovery from the illness and less infectivity. Results: trend of a faster decline in viral load with TQ formula treatment (*p* = 0.146).
To compare the percentage of RT-PCR negative/undetectable (i.e., viral clearance) on day 7 and day 14 in participants taking 3 g TQ Formula (500 mg per capsule, 3 capsules BID) versus participants taking placebo.	Percentage of negative/undetectable RT-PCR (i.e., viral clearance) on day 7 and day 14 in participants taking 3 g TQ Formula (500 mg per capsule, 3 capsules BID) versus participants taking placebo.	Treatment with 3 g TQ Formula (500 mg per capsule, 3 capsules BID) given orally on outpatient basis is hypothesized to result in higher percentages of viral clearance by RT-PCR Faster decline in viral load is hypothesized to lead to faster recovery from the illness and less infectivity. Results: Lower percentage of RT-PCR positive cases in TQ Formula arm on day 14 (24% vs. 42%, *p* = 0.22).
To compare the duration and severity of symptoms (measured by Modified FLU-PRO Plus) overtime from day 1 through day 14 in total Modified FLU-PRO Plus symptom severity score overall and in sub-domain scores (namely, Nose, Throat, Eyes, Chest/Respiratory, Gastrointestinal, Body/Systemic, Taste/Smell), between treatment with 3 g TQ Formula (500 mg per capsule, 3 capsules BID) given orally on outpatient basis and placebo in participants with COVID-19 infection.	Measurement of severity of and change in COVID-19 symptoms per total score as well as sub-scores (Nose, Throat, Eyes, Chest/Respiratory, Gastrointestinal, Body/Systemic, Taste/Smell) measured through Modified FLU-PRO Plus from day 1 through day 14 in participants with COVID-19 infection treated either with 3 g TQ Formula (500 mg per capsule, 3 capsules BID) or placebo.	FLU-PRO is a validated measure that has been used on multiple virus studies. The Modified FLU-PRO Plus version has additional questions (Taste/Smell) that are COVID-19 specific. The Modified FLU-PRO Plus has been shortened to reduce number of symptoms. Results: oral TQ Formula treated arm led to significantly faster decline in overall symptom burden from day 1 through day 14 as compared to placebo treated arm (*p* < 0.0001) as well as in throat (*p* = 0.0003), body/systemic (*p* = 0.0011), Chest/Respiratory (*p* = 0.024), and Smell/Taste (*p* = 0.032) subdomains.
To investigate if there exists an association between viral load and symptom severity by study arm and if such associations change overtime.	Correlation Coefficient of quantitative viral load and symptom severity at baseline, at day 7, and day 14 in participants taking 3 g TQ Formula (500 mg per capsule, 3 capsules BID) versus participants taking placebo	Decreases in viral load are hypothesized to be correlated with better clinical outcomes. Results: raw baseline Viral Load measure is negatively associated with total symptom burden (Spearman’s rank correlation = −0.30, *p* = 0.037). This negative rank-association does not remain significant on day 7 (*p* = 0.48) while show significance on day 14 (Spearman’s rank correlation = −0.46, *p* = 0.0031).
**Tertiary/Exploratory**		
To evaluate the basic pharmacokinetics of TQ Formula’s main active ingredient (thymoquinone) at same time points (Days 1, 7, and 14) in participants with COVID-19 infection.	Measurement of thymoquinone and metabolites’ concentration in the plasma on day 1, day 7 and 14 using HPLC in patients treated with TQ Formula.	Thymoquinone is the main active ingredient of TQ Formula and the pharmacokinetics of thymoquinone and its’ metabolites in the plasma of treated patients are used to correlate with effectiveness of treatment. Results: analysis ongoing.
To explore the effect of TQ Formula on inflammatory cytokines, coagulation factors and effector immune cells at same time points (Days 1, 7, and 14) in participants with COVID-19 infection.	Measurement of the inflammatory cytokine production, coagulation factors and the various effector immune cell subsets in the Peripheral Blood Mononuclear Cells (PBMC) of these patients on day 1, day 7 and day 14 using FACS.	The inflammatory cytokines and immunological markers are of importance because of their correlation with disease severity in COVID-19. Results: Statistically significant increases in CD4 and CD8 T-cell Percentages on day 14 (*p* = 0.042 for both).

## Data Availability

For sharing purposes, reuse conditions will be respected. The deidentified patient data will be accessible after publication of the article and can be requested from the corresponding authors (Ahmed O. Kaseb, Email: akaseb@mdanderson.org and Hassan Bencheqroun, Email h.bencheqroun@respireresearch.com) by other researchers if reuse conditions are met.

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
