# Peer review of "A Randomized, Double-Blind, Placebo-Controlled, Multicenter Study to Evaluate the Safety and Efficacy of ThymoQuinone Formula (TQF) for Treating Outpatient SARS-CoV-2"

_pathogens, 2022, doi:10.3390/pathogens11050551_

Round 1

Reviewer 1 Report

Thank you for the opportunity to review a manuscript entitled “A Randomized, Double-Blind, Placebo-Controlled, Multicenter Study to Evaluate the Safety and”. The study evaluated the safety and clinical efficacy of thymoquinone formula (TQF) through a randomized, double-blind, placebo-controlled trial. TQF appeared to be safe without any extra adverse effects compared to the placebo subgroup. A beneficial effect, yet without statistical significance, was also proved.

I have gone through the manuscript with interest, and, overall, it is very well written. The work is well described in all details. The topic is correctly introduced, methods were correctly chosen, and the findings are clearly presented and sufficiently discussed.

Few points to focus on:

I only recommend paying more attention to abbreviations and their explanation in the first use.

The discussion could confront the study with more references.

Should the manuscript be really classified as a review paper?

All in all, good job.

Author Response

please find below, also attahced file.

Few points to focus on:

I only recommend paying more attention to abbreviations and their explanation in the first use.

RESPONSE:  Thank you for taking the time to review the manuscript, we highly appreciate your comments and suggestions. Our abbreviations have been revised in the abstract page 1, line 41 and line 44, and we added another explanation for abbreviation in page 1, line 47

- The discussion could confront the study with more references.

RESPONSE:  We thank the reviewer for their comment. We revised the manuscript following your suggestion, and have now updated our references to reflect 24 cited articles. Page 12-14

Should the manuscript be really classified as a review paper?

RESPONSE:  We agree, this is a typo and we added a track change edit in the word template file to reflect clearly that this is an original research article instead of review and as the title of the manuscript states, this is “A Randomized, Double-Blind, Placebo-Controlled, Multicenter Study to Evaluate the Safety and Efficacy of ThymoQuinone Formula (TQF) for Treating outpatient SARS-CoV-2”, this is also reflected in the methods section in the abstract and in throughout the main manuscript.

Reviewer 2 Report

First of all, I would like to thank you for the opportunity to review this paper. The paper is well written and provides the necessary background for the authors' approach. The overall quality of the paper enhances the readability of the survey. While English is used generally correctly from a grammar and syntax point of view.  But this survey isn’t a  review paper.

The aim of the research has not been mentioned by the authors in the abstract, while it does not also exist in the introduction. They have to include the aim of the research in the abstract and in the introduction, as well.    Some more current literature in this domain needs to be written. The methods and techniques appear not to require further refinement.

The results are presented more than clear for the readers and the paper introduces mainly a very nice study for information

There is a very nice attempt to discuss the advantages and limitations of the current techniques which are often presented in vague terms and with well-structured references to the specifics.

I also mentioned that there are ten(10)  references at the end of the paper. But in the paper, there are more. (line 296 18-23, line 189  31,32  ).Needed to be corrected.

Author Response

First of all, I would like to thank you for the opportunity to review this paper. The paper is well written and provides the necessary background for the authors' approach. The overall quality of the paper enhances the readability of the survey. While English is used generally correctly from a grammar and syntax point of view. 

-This survey isn’t a  review paper.

-RESPONSE:  Thank you for taking the time to review the manuscript, We agree this is a typo and we added a track change edit in the word template file to reflect clearly that this is an original research article instead of review and as the title of the manuscript states, this is “A Randomized, Double-Blind, Placebo-Controlled, Multicenter Study to Evaluate the Safety and Efficacy of ThymoQuinone Formula (TQF) for Treating outpatient SARS-CoV-2”, this is also reflected in the methods section in the abstract and in throughout the main manuscript.

-The aim of the research has not been mentioned by the authors in the abstract, while it does not also exist in the introduction. They have to include the aim of the research in the abstract and in the introduction, as well.   

-RESPONSE:  We thank the reviewer for their comment. We revised the manuscript following your suggestion. We would like to point out these areas where we elude to the aim of the research in the abstract where it is now written as “Our preclinical and clinical studies aim was to determine the safety and preliminary efficacy of oral TQ Formula (TQF), in the treatment of outpatient SARS-CoV-2” in page 1, line 40-41) and in the introduction, page 2 (line 67-68) “Our preclinical and clinical studies sought to evaluate the safety and clinical efficacy of TQF through a randomized, double-blind, placebo-controlled, multicenter phase II.”

-Some more current literature in this domain needs to be written.

-RESPONSE:  We thank the reviewer for their comment. We have added a recent literature in this scope to the introduction page 2, line 64

The methods and techniques appear not to require further refinement. The results are presented more than clear for the readers and the paper introduces mainly a very nice study for information. There is a very nice attempt to discuss the advantages and limitations of the current techniques which are often presented in vague terms and with well-structured references to the specifics.

-I also mentioned that there are ten(10)  references at the end of the paper. But in the paper, there are more. (line 296 18-23, line 189  31,32  ).Needed to be corrected.

RESPONSE:  We appreciate this great suggestion very much and revised the manuscript accordingly. Some missing citations were not reflected in the reference section when formatted for the initial submission. However, they have now all been updated accordingly in pages 12-14.
